# Medication Prescribing Quality in Australian Primary Care Patients with Chronic Kidney Disease

**DOI:** 10.3390/jcm9030783

**Published:** 2020-03-13

**Authors:** Woldesellassie M. Bezabhe, Alex Kitsos, Timothy Saunder, Gregory M. Peterson, Luke R. Bereznicki, Barbara C. Wimmer, Matthew Jose, Jan Radford

**Affiliations:** School of Pharmacy and Pharmacology, University of Tasmania, Private Bage 26, Hobart, Tasmania 7001, Australia; alex.kitsos@utas.edu.au (A.K.); tim.saunder@utas.edu.au (T.S.); g.peterson@utas.edu.au (G.M.P.); luke.bereznicki@utas.edu.au (L.R.B.); barbara.wimmer@utas.edu.au (B.C.W.); Matthew.Jose@utas.edu.au (M.J.); J.Radford@utas.edu.au (J.R.)

**Keywords:** chronic kidney disease, drug therapy, quality indicators, appropriate prescribing, general practice, quality use of medicine, primary care

## Abstract

**Background:** Australian patients with chronic kidney disease (CKD) are routinely managed in general practices with multiple medications. However, no nationally representative study has evaluated the quality of prescribing in these patients. The objective of this study was to examine the quality of prescribing in patients with CKD using nationally representative primary care data obtained from the NPS MedicineWise’s dataset, MedicineInsight. **Methods:** A cross-sectional analysis of general practice data for patients aged 18 years or older with CKD was performed from 1 February 2016 to 1 June 2016. The study examined the proportion of patients with CKD who met a set of 16 published indicators in two categories: (1) potentially appropriate prescribing of antihypertensives, renin-angiotensin system (RAS) inhibitors, phosphate binders, and statins; and (2) potentially inappropriate prescribing of nephrotoxic medications, such as non-steroidal anti-inflammatory drugs (NSAIDs), at least two RAS inhibitors, triple therapy (an NSAID, a RAS inhibitor and a diuretic), high-dose digoxin, and metformin. The proportion of patients meeting each quality indicator was stratified using clinical and demographic characteristics. **Results:** A total of 44,259 patients (24,165 (54.6%) female; 25,562 (57.8%) estimated glomerular filtration (eGFR) 45–59 mL/1.73 m^2^) with CKD stages 3–5 were included. Nearly one-third of patients had diabetes and were more likely to have their blood pressure and albumin-to-creatinine ratio monitored than those without diabetes. Potentially appropriate prescribing of antihypertensives was achieved in 79.9% of hypertensive patients with CKD stages 4–5. The prescribing indicators for RAS inhibitors in patients with microalbuminuria and diabetes and in patients with macroalbuminuria were achieved in 69.9% and 62.3% of patients, respectively. Only 40.8% of patients with CKD and aged between 50 and 65 years were prescribed statin therapy. The prescribing of a RAS inhibitor plus a diuretic was less commonly achieved, with the indicator met in 20.6% for patients with microalbuminuria and diabetes and 20.4% for patients with macroalbuminuria. Potentially inappropriate prescribing of NSAIDs, metformin, and at least two RAS inhibitors were apparent in 14.3%, 14.1%, and 7.6%, respectively. Potentially inappropriate prescribing tended to be more likely in patients aged ≥65 years, living in regional or remote areas, or with socio-economic indexes for areas (SEIFA) score ≤ 3. **Conclusions:** We identified areas for possible improvement in the prescribing of RAS inhibitors and statins, as well as deprescribing of NSAIDs and metformin in Australian general practice patients with CKD.

## 1. Background

In 2015, an estimated 1.7 million Australian adults aged 18 years or older had indicators of chronic kidney disease (CKD), and of these, 604,000 had CKD stages 3–5 [1]. Diabetes and hypertension caused up to two-thirds of CKD cases. Approximately one in three and three in four Australian general practice patients with CKD had recorded diagnoses of diabetes and hypertension, respectively [2]. Progressive kidney damage with hypertension and diabetes leads to end-stage kidney disease (ESKD) [3]. In 2016 alone, 2800 new cases of ESKD were reported in Australia [4]. Patients with ESKD require expensive replacement therapy, and their treatment costs the Australian economy 1 billion per year [1]. 

Prevention of CKD progression is cost-effective and is most successful within primary care [3]. In Australia, the majority of patients with CKD stages 3–5 receive treatment from general practices [2]. Prevention of CKD progression can be achieved by treatment of modifiable risk factors and avoidance of nephrotoxic medication [3]. Kidney Health Australia’s ‘CKD management in general practice’ guideline recommends, depending on the stage of CKD, adequate treatment of hypertension, dyslipidaemia, and albuminuria [3]. It recommends controlling blood pressure at ≤140/90 mm Hg in patients with CKD alone and ≤130/80 mm Hg in those comorbid with albuminuria or diabetes [3].

Angiotensin-converting enzyme inhibitors (ACEIs) or angiotensin receptor blockers (ARBs) are first-line antihypertensive agents in CKD patients with albuminuria or diabetes [3]. These classes of drugs not only lower blood pressure but also decrease the progression of albuminuria [3]. Statin therapy is recommended in patients with CKD and aged 50 years or older as it reduces cardiovascular risk and progression of CKD [5,6]. Australian guidelines also recommend avoiding the use of medications that can potentially damage kidney function or readily accumulate and cause toxicity [6]. These medications include non-steroidal anti-inflammatory drugs (NSAID), metformin, and a high dose of digoxin [3]. 

There is limited research investigating the quality of CKD care in Australian patients [7,8,9,10]. The available studies have focused on specific classes of medications [8,9] and specific regions [7,9] or single centres [10] that may not be generalisable to the broader CKD population in Australia. There remains a need to assess prescribing quality with validated indicators in patients with CKD. Smits et al. developed a set of 16 prescribing quality indicators (PQIs) [11], which were developed according to international guidelines recommendations and are relevant to evaluate the quality of CKD care in a primary care setting [12,13]. We aimed to evaluate the quality of Australian prescribing with these indicators in patients with CKD using a large and nationally representative general practice dataset.

## 2. Methods

We analysed retrospective data obtained from the NPS MedicineWise MedicineInsight dataset. The data were de-identified and extracted from the electronic health records (EHRs) of general practices and included demographics, encounters, diagnoses, prescriptions, observations, and pathology tests. NPS MedicineWise MedicineInsight is the largest geographically representative primary care dataset in Australia. As of October 2018, NPS MedicineWise MedicineInsight had recruited 671 general practices across Australia. A total of 2,974,031 included patients had at least three clinical encounters in the previous two years. Details about this dataset can be found elsewhere [2,14,15,16]. We used MedicineInsight data collected from 329 general practices between 1 January 2013 and 1 June 2016.

In this study, we included patients with evidence of CKD based on having two renal function tests that were performed at least three months apart with: (1) estimated glomerular filtration rate (eGFR) values <60 mL/min/1.73 m^2^ or (2) albumin-to-creatinine ratio (ACR) values ≥3.5 mg/mmol for females or ≥2.5 mg/mmol for males. The renal function tests were performed between 1 January 2013 and 1 June 2015. The CKD epidemiology collaboration equation (CKD-EPI) was used to calculate eGFR [17]. This definition of CKD is congruent with that recommended for the diagnosis of CKD in Australian general practice [3]. Regular patients (defined by the Royal Australian College of General Practitioners as those with three or more encounters in the previous two years) [16] were included, if at the time of data extraction (July 2016) they were aged at least 18 years. Patients were excluded if they did not have at least one follow-up general practitioner (GP) visit between 2 June 2015 and 1 June 2016, in addition to patients who died during that period.

Variables such as age, gender, socio-economic status (based on the Index of Relative Socio-economic Advantage and Disadvantage, one of the socio-economic indexes for areas (SEIFA)) [18], rurality, continuity of care (CoC), documentation of a diagnosis of CKD, and serum electrolyte levels (e.g., calcium, and phosphate) were examined. SEIFA quintile was an index developed by the Australian Bureau of Statistics (ABS) and ranks areas in Australia from 1 (most disadvantaged area) to 5 (most advantaged area) [18]. Rurality was assigned according to the postcode of the patient’s residence and classified as major cities, regional, remote, and very remote Australia [19]. SEIFA quintile was categorised into SEIFA ≤ 3 vs. SEIFA > 3 and rurality to major cities vs. regional and remote Australia. CoC was calculated for each included patient after the time of laboratory evidence of CKD, over the remainder of the data collection period, using the Herfindahl–Hirschman Index, which has been shown to be highly correlated with other common measures of CoC [20]. Its value ranged from 0 to 1 and cut off points for low and high CoC were <0.75 and >=0.75, respectively. Low CoC in general practice, measured with this index, has also been associated with a higher risk of mortality [21,22]. 

Documentation of a diagnosis of CKD was extracted from condition codes. Baseline comorbidities, including myocardial infarction, hypertension, and diabetes were examined. The comorbidities were based on ‘condition flags’ provided by MedicineInsight. The prescribed medications that were examined included: diuretics (anatomical therapeutic chemical (ATC) code: C03), beta-blockers (C07), calcium channel blockers (C08), ACEIs (C09A), ARBs (C09C), other agents acting on renin-angiotensin system (RAS) (C09), other antihypertensives (C02), statins (C10AA or combinations as in C10BA and C10BX), phosphate binders (A12AA04, A12AA12, V03AE, and A02AB01), erythropoiesis-stimulating agents (ESAs) (B03XA), non-steroidal anti-inflammatory drugs (NSAIDs) (M01A, M01BA and B01AC), metformin (A10BA02 or in combination as A10BD), and digoxin (C01AA05). 

The recorded data included prescriptions and laboratory tests during the last four months of follow-up (between 1 February 2016 and 1 June 2016). The 16 PQIs, developed and validated by Smits et al. [11] in the Netherlands, were used to evaluate this data. They were categorised into two domains: appropriate and inappropriate PQIs. Detailed definitions for all indicators are shown in Table 4. The appropriate prescribing domain includes the first nine indicators that evaluate prescribing of all antihypertensive agents in patients with hypertension, RAS inhibitors, or diuretics in patients with proteinuria or diabetes, statins, and phosphate binders. The inappropriate prescribing domain contains the remaining seven indicators that assess the prescribing of ESA in patients with CKD and haemoglobin ≥7.5 g/dL, use of NSAIDs and metformin in patients with eGFR < 30 mL/min/1.73 m^2^, high-dose digoxin in patients with eGFR <50 mL/min/1.73 m^2^, simultaneous use of at least two RAS inhibitors, and triple therapy with an NSAID, RAS inhibitor, and diuretic. The use of phosphate binders and ESAs was not captured in our dataset as nephrologists in Australia typically prescribe them. 

Simultaneous prescribing of RAS blockers was defined as at least two of the ATC codes C09A, C09B, C09C, C09D, C09X, or combination (as in C10BX) within the last four months of the follow-up (between 1 February 2016 and 1 June 2016). Simultaneous use of NSAID, RAS blocker, and diuretic was defined as at least one prescription for each of the three classes of medications during the follow-up period. It was acknowledged that we were not capable of capturing the use of over-the-counter NSAIDs. We could also not capture whether NSAIDs were prescribed as a regular medication or for ‘as needed’ use. 

### 2.1. Statistical Analysis

Patient demographics and clinical characteristics were compared between patients with and without diabetes and are presented as numbers and proportions. The proportion of patients who met PQIs criteria are shown as percentages with 95% confidence intervals (CIs). We stratified the proportion of patients meeting each indicator by gender, age, SEIFA, rurality, CKD diagnosis documentation, and CoC. Chi-square tests were used to determine differences in the quality of prescribing with patient characteristics. All data analyses and management were conducted using the statistical and graphical computing language of R [23]. A two-sided *p* value of less than 0.05 was considered to indicate statistical significance.

### 2.2. Ethics Approval and Consent to Participate 

Ethics approval was obtained from the Tasmanian Health and Medical Human Research Ethics Committee (H0015651). De-identified data obtained from the MedicineInsight for this study did not have any patient-specific information, such as date of birth, age and postcode and individual patient consent was waived for our ethics application. Patients were informed about the programme through promotional material that was displayed with the waiting room of all participating practices. Patients choice to opt-out from the programme was respected, and robust and effective security controls safeguarded their safety.

## 3. Results

### 3.1. Baseline Charactersitics

The cohort was composed of 44,259 patients with evidence of CKD. Of these, 24,165 (54.6%) were females, and 70% were aged 70 years or older. Most patients (57.8%) had eGFR values between 45 and 59 mL/min/1.73 m^2^. Only a quarter of patients with evidence of CKD had documentation of the diagnosis, and documentation was less likely with increasing age (e.g., 51.3% for patients aged 30–39 years with evidence of CKD vs. 23.9% in those aged ≥80 years; *p* < 0.001). The sociodemographic and clinical characteristics of the study participants, including medications prescribed and monitoring performed, are shown in Table 1, Table 2 and Table 3.

A total of 13,263 patients (30%) had diabetes. Of these, 11,608 (87.5%) and 6608 (87.5%) had hypertension and a history of myocardial infarction, respectively (Table 2). Of 39,716 (89.7%) patients who had a recorded blood pressure measurement, 13,338 (33.6%) had uncontrolled blood pressure (>140/90 mm Hg). The proportion of patients with uncontrolled blood pressure was slightly higher in patients with diabetes (34.4% vs. 33.2%, *p* = 0.03) than in those without diabetes. Antihypertensive medication prescribing was significantly higher in CKD patients with diabetes compared with those without diabetes (82.1% vs. 70.6%, *p* < 0.001). Compared with CKD patients without diabetes, CKD patients with diabetes were more likely to be prescribed an ACEI/ARB (64.1% vs. 51.5%, *p* < 0.001). Over 60% of CKD patients with diabetes were prescribed a statin compared with less than 40% without diabetes (*p* < 0.001) (Table 3).

Only a few patients had recorded treatment with phosphate binders, ESAs, and vitamin D. Therefore, the three PQIs: seven, eight, and nine that assess appropriate prescribing of phosphate binders and the two PQIs: 11 and 12 that evaluate inappropriate prescribing of vitamin D and ESAs were not operational. These five indicators were excluded from further analyses. 

### 3.2. Appropriate Prescribing 

Among patients with CKD stages 4–5 and hypertension, 79.9% overall and 83.5% of those aged ≥65 years were prescribed antihypertensive agents. The proportion of patients with microalbuminuria (ACR 2.5–25 mg/mmol for males, 3.5–35 mg/mmol for females) and diabetes who were prescribed an ACEI/ARB or an ACEI/ARB plus a diuretic were 69.9% and 20.6%, respectively. Overall, the prescribing of ACEI/ARB in patients with macroalbuminuria (ACR > 25 mg/mmol for males, >35 mg/mmol for females) was 62.3%. This was significantly higher in those patients aged ≥65 years than those < 65 years (65.4% vs. 56.1%, *p* < 0.001) and in those without documented CKD diagnosis (64.5% vs. 60.0%, *p* = 0.046) than those documented. The proportion of patients with macroalbuminuria who were prescribed an ACEI/ARB plus a diuretic was 20.4%, overall, and was significantly higher in those aged ≥65 years (22% vs. 16.7%, *p* = 0.021) than those <65 years (Table 4 and Appendix A). 

We examined the prescribing of statins in CKD patients with diabetes and in those aged between 50 and 65 years, as guidelines recommend statin use in both of these groups [12]. The proportion of patients who were prescribed a statin was 39.9% in patients without diabetes and 60.6% in patients with diabetes. The percentage of statin prescribing was 40.8% in patients with CKD aged between 50 and 65 years. Prescribing of statins in this age group was more common in patients with a SEIFA score ≤3 than >3 (45.3% vs. 38.9%, *p* < 0.001) and in patients with a documented CKD diagnosis (45.1% vs. 38.9%, *p* < 0.001) (Table 4 and Appendix A). 

### 3.3. Potentially Inappropriate Prescribing 

The percentage of patients with potentially inappropriate prescribing of an NSAID in combination with a RAS blocker and a diuretic (triple therapy) was 2.6%, overall. It was higher in those whose CKD diagnosis was documented (3.0% vs. 2.5%, *p* = 0.002) than not documented, and in those aged ≥65 years (2.7% vs. 2.0%, *p* = 0.004) than aged <65 years. It was slightly higher in patients with SEIFA score ≤3 than >3 (3.2% vs. 2.4%, *p* < 0.001) and in CKD patients living in regional and remote areas than in patients living in major cities (2.8% vs. 2.5%; *p* = 0.032) (Table 4 and Appendix A). Among patients with eGFR < 30 mL/min/1.73 m^2^, the proportion prescribed an NSAID was 14.3% overall and was higher in patients aged ≥65 years (15.1 vs. 9.4%, *p* < 0.001) than those aged <65 years and in patients with SEIFA score ≤3 (15.9% vs. 13.6%, *p* = 0.033) than those with SEIFA score >3. 

Of those patients with CKD stages 3–5 and prescribed a RAS blocker, 7.6% were prescribed at least two RAS blockers simultaneously. This was more likely in patients with SEIFA score ≤3 than >3 (8.3% vs. 7.3%; *p* = 0.005) and in patients living in major cities than those living in regional and remote areas (8.0% vs. 7.0%; *p* = 0.002) (Table 4 and Appendix A).

There were 5130 patients with diabetes who were prescribed metformin. Of 1967 patients with a diagnosis of diabetes and with an eGFR < 30 mL/min/1.73 m^2^, 278 (14.1%) were potentially inappropriately prescribed metformin. This was slightly greater in patients living in regional and remote Australia (16.8%) than those living in major cities (12.3%; *p* = 0.005) and in patients whose CKD diagnosis was not documented (16.3% vs. 12.5%; *p* = 0.018) (Table 4, Appendix A). 

In patients with an eGFR < 50 mL/min/1.73 m^2^, the proportion prescribed high-dose digoxin (0.125 mg/day) was 3.8%. This was higher in females (4.1% vs. 3.3%, *p* < 0.001), in those aged ≥65 years (4.0% vs. 1.1%, *p* < 0.001) than aged <65 years, and in those living in regional and remote areas (4.2% vs. 3.5%, *p* = 0.002) than those living in major cities (Table 4 and Appendix A). 

## 4. Discussion

This study is the most extensive to date that evaluates the quality of medication prescribing in Australian general practice patients with CKD, utilising a set of 16 validated indicators and based on diabetes status [11]. Potential gaps in prescribing CKD progression protective medications and avoiding nephrotoxic drugs were identified. ACEIs/ARBs in patients with proteinuria or diabetes and statins in patients aged between 50 and 65 years were found to be under-prescribed. Potential inappropriate prescribing identified included simultaneous prescribing of at least two RAS inhibitors, prescribing of NSAIDs and metformin in patients with eGFR <30 mL/min/1.73 m^2^, and use of high-dose digoxin in patients with an eGFR < 50 mL/min/1.73 m^2^. With at least one indicator, inappropriate prescribing was more common in patients with SEIFA ≤ 3, aged ≥65 years, or living in regional and remote Australia. Compared with patients without diabetes, patients with diabetes generally received more comprehensive blood pressure and laboratory monitoring and pharmacotherapy. 

Despite strong evidence for the efficacy of ACEI/ARB to reduce proteinuria and slow progression of CKD to ESKD, less than 70% of Australian adult patients with CKD stages 3–5 with diabetes and microalbuminuria were receiving an ACEI or ARB. The prescribing of an ACEI or ARB in patients with CKD with albuminuria was slightly lower in Australian general practice compared to that reported in other developed nations [24,25,26,27]. Studies from different provinces of Canada [24,25,26] investigating prescribing in CKD patients reported rates of 74% to 80% for ACEI or ARB prescribing, while a study conducted in the Netherlands found prescribing in 78% and 82% of non-diabetes and diabetes patients, respectively [11]. The reason for the low rate of ACEI/ARB prescribing could be non-concordance to Australian CKD treatment guidelines, including not referring patients to nephrology care [3]. The cost of ACEI/ARB probably had a limited impact on the rate of their prescription as these medications are subsidized by the Australian Pharmaceutical Benefits Scheme (PBS). 

It was unexpected to find no difference or even low rates of an ACEI or ARB/an ACEI or ARB plus diuretic prescribing in patients with proteinuria and documented CKD compared to those without documented CKD. This might suggest that GPs awareness of patients’ CKD status did not necessarily compel them to prescribe an ACEI/ARB. They may have other valid reasons for not prescribing, including hyperkalemia, hypotension, and acute renal injury (AKI) [28]. Among CKD patients with proteinuria who were receiving multiple antihypertensive agents, only a fifth use an ACEI/ARB in combination with a diuretic (double therapy) in this study. Double and triple (ACEI/ARB plus a diuretic plus an NSAID) therapies are associated with AKI [29], which might have discouraged GPs from prescribing. However, compared with those with triple therapy, the risk of developing AKI is less likely in patients with double therapy [29]. Combining ACEI/ARB with a thiazide diuretic instead of a loop diuretic might reduce the risk of discontinuation of ACEI/ARB [30]. The risk of inducing hypotension and the associated fall in elderly patients might outweigh the renoprotective effect gained by combining an ACEI/ARB with a diuretic, and it might have also prevented GPs from prescribing [30,31]. 

Statins are relatively well-tolerated medications and are beneficial in lowering the risk of cardiovascular events in patients with CKD [12,32]. Notwithstanding the PBS restrictions on the prescribing of statins, the current Kidney Disease: Improving Global Outcomes (KDIGO) and Kidney Health Australia’s guidelines [3,12] recommend statin or statin/ezetimibe treatment in adults aged 50 years and over with eGFR < 60 mL/min/1.73 m^2^ but not treated with chronic dialysis or kidney transplantation. In this study, only 40.8% of patients aged 50 to 65 years were receiving statins. These rates were less than a 54% lipid-lowering medication prescribing rate in primary care patients with CKD reported by a prior Australian study, AusHEART [33]. A study by Smits et al. [11] in the Netherlands using the same indicators reported a higher (74%) rate of statin prescribing in patients with CKD stages 3–5 aged 50 to 65 years. The significantly low rate of statin prescribing in those patients without documented CKD diagnosis (38.9%) suggests that lack of CKD recognition by GPs might be one reason for the low rate of statin prescribing.

Statin side effects and interactions were the main concerns of Australian patients taking statin [34], and which were also cited as the most commons reasons for statin discontinuation elsewhere [35]. The low rate of statin therapy in this study could be related to public concern over perceived adverse reactions following an extensive media campaign about the negative effect of statins [36]. The other likely reason is the lack of PBS subsidisation for statin therapy for CKD in the absence of other indications [37]. 

Our study indicated that patients living in relatively disadvantaged socio-economic areas (SEIFA score ≤ 3) were more likely to be prescribed potentially inappropriate NSAIDs, simultaneous use of at least two RAS inhibitors, and triple therapy (combined use of an NSAID, a RAS inhibitor and a diuretic). Similarly, patients from regional or remote areas of Australia were more likely to be prescribed potentially inappropriate digoxin, metformin, and triple therapy. These findings suggest that patients living in disadvantaged socio-economic areas, as well as regional and remote areas, may receive a lower quality of CKD care than patients living in socio-economic most advantaged areas or major cities. The health care inequality between regional/remote areas and major cities of Australia has been the subject of many reports and initiatives [38]. Three in five people in remote/very remote areas did not see a specialist because of distance, and people in outer regional and remote/very remote areas were less likely to have a usual GP [38]. Inequality in prescribing has also been found elsewhere. A study in France [39] reported that inappropriate prescribing was highest in older people living in municipalities with low socio-economic status. A similar study conducted in Ireland [40] also found that inappropriate prescribing was more prevalent in relatively deprived patients aged over 70 years.

In this study, patients with CKD and aged 65 years or over were more likely to be prescribed nephrotoxic medications: triple therapy, high-dose digoxin, and NSAIDs. One possible explanation is that some GPs may not consider an eGFR measurement between 45–59 mL/min/1.73 m^2^ as evidence of CKD in older individuals. They might consider these eGFR values as reflecting the normal physiological changes related to aging. 

Unlike a previous study by Khanam et al. [31], using MedicineInsight data, which found higher CoC led to better blood pressure control, this study found no significant differences in prescribing quality between patients with higher and lower CoC.

## 5. Strengths and Limitations

This study had a large sample size, and patient characteristics within the MedicineInsight dataset are similar to the Australian population [2,14,16]. However, there are several limitations. Medications prescribed solely by specialists, such as nephrologists and cardiologists, who worked in hospitals and speciality clinics were not recorded in NPS MedicineWise MedicineInsight. For instance, phosphate binders and ESAs are not usually prescribed by GPs (generally prescribed by nephrologists), and thus our data were not complete on the use of these medications.

We did not account for medication contraindications and adverse drug reactions that may have prevented GPs from prescribing a specific class of medication to patients. Adverse drug reactions are recorded in free text in ‘Allergies/Reactions Table’ in the NPS MedicineWise MedicineInsight dataset. This table is not an event-based table and does not necessarily record each occurrence of adverse drug reaction. Free-text search for an adverse drug reaction from this table is of poor quality.

NSAIDs are also available without a prescription, but we could only obtain data on prescribed NSAIDs. Simultaneous prescribing of at least two RAS blockers within the four months might not necessarily indicate concomitant inappropriate use. It might be an overlapping period of switching from one RAS blocker monotherapy to the other. We also did not investigate the impact of medication use on patient outcomes.

GPs collected the data for clinical decision making, not for research purposes. The EHRs may not contain all sociodemographic and clinical characteristics. For instance, indigenous status was not recorded for 24.3% of the patients. There is a possibility that aspects of patients’ medical history, prescriptions, and laboratory tests were recorded in notes and not included in the research data, which used specified fields and not the body of free-text consultation notes.

We noted that including only regular patients (who had three or more clinical encounters in past two years) in this study potentially introduced selection bias by including more older patients with multiple comorbidities who visited their GP more frequently. However, four in five Australian patients visited their GP multiple times in a year [41], and nearly all patients could visit their GP at least three times in two years. In conclusion, we identified the potential for possible improvement in the prescribing of recommended preventive medications and deprescribing of nephrotoxic medication in patients with CKD in Australian primary care. Programmes to optimise the quality use of medications should focus on improving the prescribing practices for protective medications, such as an ACEI or ARB and a statin, and deprescribing concurrent NSAIDs and RAS blockers in patients with CKD.

## Figures and Tables

**Table 1 jcm-09-00783-t001:** Baseline sociodemographic characteristics of patients with chronic kidney disease (CKD) overall and by diabetes status.

	Overall, n = 44,259 n (%)	Diabetes		*p* Value
		No n = 30,996 n (%)	Yes n = 13,263 n (%)	
Age groups (years)				<0.836
<65	4373 (9.9)	3069 (9.9)	1304 (9.8)	
≥65	39,886 (90.1)	27,927 (90.1)	11,959 (90.2)	
Female	24,165 (54.6)	17,620 (56.8)	6545 (49.3)	<0.001
Indigenous Status				<0.001
Indigenous	436 (1.0)	212 (0.7)	224 (1.7)	
Non-Indigenous	33,067 (74.7)	23,020 (74.3)	10,047 (75.8)	
Missing	10,756 (24.3)	7764 (25.0)	2992 (22.6)	
SEIFA quintile *				
≤3	12,254 (27.8)	8302 (26.9)	3952 (30.0)	<0.001
>3	31,754 (72.2)	22,559 (73.1)	9225 (70.0)	
Missing	251 (0.6)	165 (0.5)	86 (0.6)	
Rurality *				<0.001
Major Cities of Australia	26,617 (60.4)	18,468 (59.9)	8149 (61.8)	
Regional and Remote Australia	17,420 (39.6)	12,385 (40.1)	5035 (38.2)	
Missing	222 (0.5)	143 (0.5)	79 (0.6)	
GP Continuity of Care				<0.001
Low (<0.75)	17,421 (39.4)	11,917 (38.5)	5504 (41.5)	
High (≥0.75)	26,833 (60.6)	19,075 (61.5)	7758 (58.5)	
Missing	5 (0.0)	1 (0.0)	4 (0.0)	
Documentation of CKD	11,618 (26.3)	7722 (24.9)	3896 (29.4)	<0.001

SEIFA, socio-economic indexes for areas; GP, general practitioner. * Excludes patients without a recorded postcode in the electronic health record.

**Table 2 jcm-09-00783-t002:** Comorbidities of patients with CKD overall and by diabetes status.

	Overall, n = 44,259 n (%)	Diabetes		*p* Value
		No n = 30,996 n (%)	Yes n = 13,263 n (%)	
CKD Stage				<0.001
Stage 3a (45–59 mL/min/1.73 m^2^)	25,562 (57.8)	18,724 (60.4)	6838 (51.6)	
Stage 3b (30–44 mL/min/1.73 m^2^)	13,551 (30.6)	9093 (29.3)	4458 (33.6)	
Stage 4 (15–29 mL/min/1.73 m^2^)	4186 (9.5)	2573 (8.3)	1613 (12.2)	
Stage 5 (<15 mL/min/1.73 m^2^)	960 (2.2)	606 (2.0)	354 (2.7)	
ACR (mg/mmol)				0.023
Normal <2.5 (male) <3.5 (female)	7877 (17.8)	3838 (12.4)	4039 (30.5)	
Microalbuminuria 2.5–25 (male) 3.5–35 (female)	4707 (10.6)	1719 (5.6)	2978 (22.5)	
Macroalbuminuria >25 (male) >35 (female)	2427 (5.5)	897 (2.9)	1530 (11.5)	
Missing	29,248 (66.1)	24,532 (79.1)	4716 (35.6)	
Indigenous Status				<0.001
Indigenous	436 (1.0)	212 (0.7)	224 (1.7)	
Non-Indigenous	33,067 (74.7)	23,020 (74.3)	10,047 (75.8)	
Missing	10,756 (24.3)	7764 (25.0)	2992 (22.6)	
Comorbidities				
Hypertension	35,386 (80.0)	23,778 (76.7)	11,608 (87.5)	<0.001
Myocardial infarction	17,945 (40.5)	11,688 (37.7)	6257 (47.2)	<0.001
Atrial fibrillation	7038 (15.9)	4893 (15.8)	2145 (16.2)	0.315
Anxiety	5658 (12.8)	4124 (13.3)	1534 (11.6)	<0.001
Bipolar disorder	505 (1.1)	365 (1.2)	140 (1.1)	0.290
Schizophrenia	363 (0.8)	227 (0.7)	136 (1.0)	0.002

ACR, albumin-to-creatinine ratio; CKD, chronic kidney disease.

**Table 3 jcm-09-00783-t003:** Proportion of patients with CKD receiving monitoring and medications by diabetes status.

	Total n = 44,259 n (%)	Diabetes		*p* Value
		No n = 30,996 n (%)	Yes n = 13,263 n (%)	
Blood Pressure				
Patients with BP Recorded	39,716 (89.7)	27,411 (88.4)	12,305 (92.8)	<0.001
Low Diastolic BP (<70 mmHg)	13,602 (34.2)	8935 (32.6)	4667 (37.9)	<0.001
High Systolic BP (>140 mmHg)	13,338 (33.6)	9108 (33.2)	4230 (34.4)	0.033
Pathology				
Patients with phosphate test recorded	23,133 (52.3)	16,060 (51.8)	7073 (53.3)	0.004
Elevated phosphate level (>1.49 mmol/L)	1322 (5.7)	872 (5.4)	450 (6.4)	0.005
Patients with calcium test recorded	22,818 (51.6)	16,096 (51.9)	6722 (50.7)	0.017
Elevated calcium level (>2.54 mmol/L)	1343 (5.9)	893 (5.5)	450 (6.7)	<0.001
Low calcium level (<2.10 mmol/L)	589 (2.6)	409 (2.5)	180 (2.7)	0.584
Patients with Hb test recorded	40,601 (91.7)	28,723 (92.7)	11,878 (89.6)	<0.001
Low Hb level (<7.5 mmol/L)	14,125 (34.8)	9252 (32.2)	4873 (41.0)	<0.001
**Medication**				
Antihypertensives				
At least one antihypertensives ^≠^	32,782 (74.1)	21,893 (70.6)	10,889 (82.1)	<0.001
Diuretic	9539 (21.6)	5956 (19.2)	3583 (27.0)	<0.001
Beta Blocker	10,763 (24.3)	6862 (22.1)	3901 (29.4)	<0.001
Calcium Channel Blocker	9551 (21.6)	6232 (20.1)	3319 (25.0)	<0.001
ACEI or ARB	24,485 (55.3)	15,978 (51.5)	8507 (64.1)	<0.001
Multiple ACEI or ARB	1859 (4.2)	1066 (3.4)	793 (6.0)	<0.001
Statin	20,411 (46.1)	12,370 (39.9)	8041 (60.6)	<0.001
All phosphate binders	244 (0.6)	155 (0.5)	89 (0.7)	0.031
Non-calcium-containing phosphate binders	67 (0.2)	41 (0.1)	26 (0.2)	0.148
Calcium-containing phosphate binders	182 (0.4)	119 (0.4)	63 (0.5)	0.197
Vitamin D	1444 (3.3)	939 (3.0)	505 (3.8)	<0.001
ESAs	42 (0.1)	24 (0.1)	18 (0.1)	0.098
NSAIDs	7426 (16.8)	4862 (15.7)	2564 (19.3)	<0.001
Metformin	5189 (11.7)	59 * (0.2)	5130 (38.7)	<0.001
Digoxin	1516 (3.4)	976 (3.1)	540 (4.1)	<0.001

BP, blood pressure; Hb, haemoglobin; ACEI, angiotensin-converting enzyme inhibitor; ARB, angiotensin receptor blocker; ESAs, erythropoiesis-stimulating agents; NSAIDs, non-steroidal anti-inflammatory drugs. ^≠^ Includes all antihypertensives with anatomical therapeutic chemical (ATC) code C02, C03, C07, C08, C09, or combinations (as in C10BX). * Patients with a prescription for metformin who did not have a recorded diagnosis of type 2 diabetes.

**Table 4 jcm-09-00783-t004:** Number and proportion of patients meeting prescribing quality indicators by rurality, socio-economic indexes for areas (SEIFA) and CKD documentation [4].

Quality Indicator			Numerator	Denominator	Percentage	*p* Value
Treatment of Hypertension						
1. Percentage of patients aged 18 to 80 years with CKD stages 4–5 and hypertension who are prescribed antihypertensives unless undesirable because of low diastolic blood pressure	Overall *		1029	1288	79.9	
	Rurality	Major cities of Australia	532	672	79.2	0.565
		Regional and Remote Australia	490	609	80.5	
	SEIFA quintile	≤3	345	433	79.7	0.947
		>3	677	848	79.8	
	CoC	High	375	459	81.7	0.228
		Low	654	829	78.9	
	CKD documented	No	380	485	78.4	0.284
		Yes	649	803	80.8	
	Systolic BP	>140 mmHg	455	573	79.4	0.541
		≤140 mmHg	588	728	80.8	
	Age	<65 years	318	437	72.8	<0.001
		≥65 years	711	851	83.5	
	Sex	Female	450	561	80.2	0.800
		Male	579	727	79.6	
2. Percentage of patients aged 18 to 80 years with CKD stages 3–5 and macroalbuminuria treated with multiple antihypertensives who are prescribed a combination of an ACEI or ARB and a diuretic	Overall *		298	1464	20.4	
	Rurality	Major cities of Australia	174	837	20.8	0.679
		Regional and Remote Australia	123	618	19.9	
	SEIFA quintile	≤3	94	496	19.0	0.315
		>3	203	958	21.2	
	CoC	High	104	528	19.7	0.639
		Low	194	936	20.7	
	CKD documented	No	148	751	19.7	0.527
		Yes	150	713	21.0	
	Systolic BP	>140 mmHg	143	643	22.2	0.123
		≤140 mmHg	150	792	19.0	
	Age	<65 years	74	444	16.7	0.021
		≥65 years	224	1020	22	
	Sex	Female	102	468	21.8	0.348
		Male	196	996	19.7	
3. Percentage of patients aged 18 to 80 years with CKD stages 3–5, microalbuminuria and diabetes treated with multiple antihypertensives who are prescribed a combination of an ACEI or ARB and a diuretic	Overall *		337	1634	20.6	
	Rurality	Major cities of Australia	190	956	19.9	0.270
		Regional and Remote Australia	147	664	22.1	
	SEIFA quintile	≤3	110	513	21.4	0.672
		>3	227	1106	20.5	
	CoC	High	144	641	22.5	0.140
		Low	193	993	19.4	
	CKD documented	No	216	1075	20.1	0.462
		Yes	121	556	21.8	
	Systolic BP	>140 mmHg	119	563	21.1	0.667
		≤140 mmHg	213	1053	20.2	
	Age	<65 years	40	228	17.5	0.215
		≥65 years	297	1406	21.1	
	Sex	Female	149	655	22.7	0.083
		Male	188	979	19.2	
Treatment of albuminuria						
4. Percentage of patients aged 18 to 80 years with CKD stages 3–5 and macroalbuminuria who are prescribed an ACEI or ARB	Overall *		1084	1741	62.3	
	Rurality	Major cities of Australia	636	1016	62.6	0.725
		Regional and Remote Australia	441	714	61.8	
	SEIFA quintile	≤3	353	573	61.6	0.705
		>3	723	1156	62.5	
	CoC	High	387	645	60.0	0.135
		Low	697	1096	63.6	
	CKD documented	No	578	898	64.5	0.046
		Yes	506	845	60.0	
	Age	<65 years	331	590	56.1	<0.001
		≥65 years	753	1151	65.4	
	Sex	Female	327	544	60.1	0.212
		Male	757	1197	63.2	
5. Percentage of patients aged 18 to 80 years with CKD stages 3–5, microalbuminuria and diabetes who are prescribed an ACEI or ARB	Overall *		1252	1790	69.9	
	Rurality	Major cities of Australia	738	1064	69.4	0.516
		Regional and Remote Australia	502	709	70.8	
	SEIFA quintile	≤3	393	546	72.0	0.207
		>3	846	1226	69.0	
	CoC	High	502	705	71.2	0.348
		Low	750	1085	69.1	
	CKD documented	No	841	1179	71.3	0.075
		Yes	411	611	67.3	
	Age	<65 years	176	259	68	0.450
		≥65 years	1076	1531	70.3	
	Sex	Female	496	711	69.8	0.891
		Male	759	1079	70.1	
Prescription of statins						
6. Percentage of patients aged 50 to 65 years with CKD stages 3–5 who are prescribed a statin	Overall *		1508	3693	40.8	
	Rurality	Major cities of Australia	823	2023	40.7	0.898
		Regional and Remote Australia	669	1636	40.9	
	SEIFA quintile	≤3	488	1077	45.3	<0.001
		>3	1004	2581	38.9	
	CoC	High	542	1292	42.0	0.311
		Low	966	2401	40.2	
	CKD documented	No	991	2547	38.9	<0.001
		Yes	517	1146	45.1	
	Sex	Female	714	1814	39.4	0.073
		Male	794	1879	42.3	
Treatment of MBD						
7. Percentage of patients aged 18 to 80 years with CKD stages 3–5 and with an elevated phosphate level who are prescribed a phosphate binder			54	815	6.6	
8. Percentage of patients aged 18 to 80 years with CKD stages 3–5 treated with phosphate binders and with an elevated calcium level who are prescribed a non-calcium-containing phosphate binder			5	7	71.4	
9. Percentage of patients aged 18 to 80 years with CKD stages 3–5 treated with phosphate binders and with a low calcium level who are prescribed a calcium-containing phosphate binder			6	12	50.0	
Medication safety						
10. Percentage of patients 18 years or older with CKD stages 3–5 and a prescription of RAS blockers who are prescribed at least two RAS blockers simultaneously	Overall *		1859	24,485	7.6	
	Rurality	Major cities of Australia	1175	14,639	8.0	0.002
		Regional and Remote Australia	678	9732	7.0	
	SEIFA quintile	≤3	587	7037	8.3	0.005
		>3	1264	17,317	7.3	
	CoC	High	727	9756	7.5	0.499
		Low	1132	14,729	7.7	
	CKD documented	No	1339	18,073	7.4	0.069
		Yes	520	6412	8.1	
	Age	<65 years	182	2076	8.8	0.035
		≥65 years	1677	22,409	7.5	
	Sex	Female	1007	13,466	7.5	0.455
		Male	852	11,019	7.7	
11. Percentage of patients 18 years or older with CKD stages 3–5 and elevated calcium levels who are prescribed active vitamin D			67	1343	5.0	
12. Percentage of patients 18 years or older with CKD stages 3–5 and Hb ≥ 7.5 who are prescribed ESA			0	26,476	0.0	
13. Percentage of patients 18 years or older with Egfr < 30 mL/min/1.73 m^2^ who are prescribed a NSAID	Overall *		735	5146	14.3	
	Rurality	Major cities of Australia	421	3054	13.8	0.201
		Regional and Remote Australia	312	2072	15.1	
	SEIFA quintile	≤3	238	1496	15.9	0.033
		>3	494	3627	13.6	
	CoC	High	295	2058	14.3	0.935
		Low	440	3087	14.3	
	CKD documented	No	352	2367	14.9	0.266
		Yes	383	2779	13.8	
	Age	<65 years	68	724	9.4	<0.001
		≥65 years	667	4422	15.1	
	Sex	Female	360	2648	13.6	0.147
		Male	375	2498	15.0	
14. Percentage of patients 18 years or older with Egfr < 30 mL/min/1.73 m^2^ and diabetes who are prescribed metformin	Overall ^*^		278	1967	14.1	
	Rurality	Major cities of Australia	149	1208	12.3	0.005
		Regional and Remote Australia	126	749	16.8	
	SEIFA quintile	≤3	80	608	13.2	0.444
		>3	195	1349	14.4	
	CoC	High	119	829	14.4	0.810
		Low	159	1138	14.0	
	CKD documented	No	136	835	16.3	0.019
		Yes	142	1132	12.5	
	Age	<65 years	32	244	13.1	0.625
		≥65 years	246	1723	14.3	
	Sex	Female	127	957	13.3	0.285
		Male	151	1010	15	
15. Percentage of patients 18 years or older with eGFR < 50 mL/min/1.73 m^2^ who are prescribed digoxin > 0.125 mg/day	Overall *		995	26,434	3.8	
	Rurality	Major cities of Australia	558	16,020	3.5	0.002
		Regional and Remote Australia	433	10,282	4.2	
	SEIFA quintile	≤3	293	7394	4.0	0.295
		>3	697	18,893	3.7	
	CoC	High	366	10,623	3.4	0.025
		Low	629	15,807	4.0	
	CKD documented	No	696	17,547	4.0	0.015
		Yes	299	8887	3.4	
	Age	<65 years	25	2252	1.1	<0.001
		≥65 years	970	24,182	4.0	
	Sex	Female	596	14,411	4.1	<0.001
		Male	399	12,023	3.3	
16. Percentage of patients 18 years or older with CKD stages 3–5 and who are prescribed with a combination of NSAID, RAS blocker and diuretic	Overall *		1160	44,259	2.6	
	Rurality	Major cities of Australia	663	26,617	2.5	0.032
		Regional and Remote Australia	492	17,420	2.8	
	SEIFA quintile	≤3	397	12,254	3.2	<0.001
		>3	757	31,758	2.4	
	CoC	High	452	17,421	2.6	0.777
		Low	708	26,833	2.6	
	CKD documented	No	809	32,641	2.5	0.002
		Yes	351	11,618	3.0	
	Age	<65 years	86	4373	2.0	0.004
		≥65 years	1074	39,886	2.7	
	Sex	Female	640	24,165	2.6	0.691
		Male	520	20,094	2.6	

BP, blood pressure; CKD, chronic kidney disease; MBD, mineral and bone density; CoC, continuity of care; SEIFA, socio-economic indexes for areas; ACEI, angiotensin-converting enzyme inhibitor; ARB, angiotensin receptor blocker; RAS, renin-angiotensin system; eGFR, estimated glomerular filtration rate; ESAs, erythropoiesis-stimulating agents; NSAIDs, non-steroidal anti-inflammatory drugs. * ‘Patient SEIFA’, ‘Patient Rurality’, Patient CoC’, and ‘CKD documented’ for the indicator does not add up to ‘Overall’ due to missing data.

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
