# Peer review of "Medication Prescribing Quality in Australian Primary Care Patients with Chronic Kidney Disease"

_jcm, 2020, doi:10.3390/jcm9030783_

Round 1
Reviewer 1 Report
This manuscript describes prescribing practices among Australian GP's for patients with documented or undocumented CKD using QI criteria developed by Smits et al. The large number of participating GP's and consequently patients and the prescribing central database are definite strengths of the paper and I recommend its publication.
A few suggestions:
1. For those readers unaware of practice patterns in Australia, I would mention the limitation that nephrologists prescribe PO4 binders and ESA's earlier in the text. My initial reading and examination of Table 3 raised a question about why those criteria were not examined.
2. Several times in the text (ex line 67) the authors mention that RAS inhibitors prevent CKD progression. I wish they did. There is excellent evidence that they retard progression but do not prevent it. I suggest modifying those statements.
3. I do not understand the rational/need for p values in Table 1.
4. There are a few typos that a spellchecker will identify.
Reviewer 2 Report
This is a mainly descriptive article about medication prescription in stage 1 to 5 CKD patients in Australian general practice. It focuses on the frequency of appropriate drug prescription that is in line with national and international CKD guidelines, versus the frequency of inappropriate or harmful drug prescription. It also describes differences in drug prescription across CKD stages, albuminuria stages, in diabetics vs non-diabetics, and across various socioeconomic and geographical groups. The article also presents some data on achieved blood pressure levels and metabolic status in these patients.
The main strengths of the article is the high number of patients included, and the rich description of comorbidities, lab values, blood pressure, and medication classes. Its main weaknesses are the discussion section with little added analytical value beyond description, the unclear presentation with excessively large tables, and no clear statement of how or if consent was obtained from the patients.
Major concerns:
- Discussion section: There is little added analytical value to the discussion beyond descriptions. It would be interesting to hear the authors elaborate further on why treatment rates with ACEi/ARBs were not higher in CKD patients, and in fact lower in albuminuric CKD patients than in albuminuric patients without CKD. Does drug price play a role here? Or the fear of acute kidney injury in an elderly population? Or the low availability of nephrologist care? Statin prescription is more elaborately discussed, but what about the role of statin intolerance/side-effects, drug interactions, "bad publicity" etc in explaining low prescription rates? The discussion section adds little novelty or insight to these issues. Also, possible selection biases are not adressed. For example, documentation of CKD diagnoses is based on doctor-assigned condition codes rather than lab-based eGFR. Furthermore, including only patients with 3 or more GP encounters in the given time frame could possibly introduce selection bias towards older and more multimorbid patients, and a discussion of this, with thoughts on possible non-visiting CKD patients, should be included.
- Missing data: There is no presentation of missing data in the tables or in the text. When some 32,700 patients use antihypertensives out of 44,200 patients, does it mean that almost 12,000 CKD patients use no antihypertensives, or are the data incomplete? Missing data are important in large epidemiological studies, and should be accounted for.
- Presentation: Both table 1, 2 and 3 are very long and dense with information. It could perhaps be more informative if table 1 was split into two tables (e.g. 1a with CKD/comorbidities, and 1b with geographical and epidemiological variables). Table 3 could be moved to a supplementary section entirely. Furthermore, Figures 1 and 2 in the Supplementary data would be more readily legible if the PQI variables were tagged with simple key words (such as antihypertensive meds, ACEi/ARB in macroalbuminuria, statin use etc), rather than with the more difficult-to-understand PQI-1, PQI-2 etc. As it stands now, at least table 3 is impenetrable.
- Lack of consent: The data were cropped from electronic health records, but there is no proper description of how or if consent was obtained from the patients. I am sure that the study was conducted in an ethically sound way. The paper, however, does not explain what kind of active or assumed consent patients have given to allow their clinical GP data to be published (albeit anonymized).
Minor concerns:
- Diabetes vs non-diabetes: Patient demographics and clinical characteristics were presented in a dichotomized way, with diabetics vs non-diabetics. Although biologically interesting, this focus on diabetes vs non-diabetes was not prespecified in the title, abstract or introduction, and does not carry into the discussion. It seems reasonable to either focus more on diabetes vs non-diabetes, or drop it entirely, so that the other scientific points can come to light.
- Urine tests: The article does not say how many urine tests are analyzed to classify the degree of proteinuria for each patient. KDIGO recommends two or more urine tests. This is not clearly stated here.
All in all, the paper addresses an interesting topic that is central to the quality of care of the many CKD patients in Australia. However, it is not fully able to present its findings in an easily comprehensible way, or discuss the findings in an analytical way that adds scientific value to the descriptions.
Author Response
Response to Reviewers’ file
Manuscript: jcm-723275
“Medication Prescribing quality in Australian primary care patients with chronic kidney disease”
Authors: Woldesellassie M. Bezabhe, Alex Kitsos, Timothy Saunder, Gregory M. Peterson, Luke R. Bereznicki, Matthew Jose, Barbara Wimmer, Jan Radford
Dear Dr. Edwin Zhang,
Editor-in-Chief,
Journal of Clinical Medicine
Dear Dr. Edwin Zhang,
We would like to extend our deepest gratitude to both editors and reviewers, for giving us such constructive comments. We have revised the manuscript based on points raised by the academic editors and reviewers. Our point-by-point amendments and responses are as shown below in italics and highlighted style.
Reviewer 2
This is a mainly descriptive article about medication prescription in stage 1 to 5 CKD patients in Australian general practice. It focuses on the frequency of appropriate drug prescription that is in line with national and international CKD guidelines, versus the frequency of inappropriate or harmful drug prescription. It also describes differences in drug prescription across CKD stages, albuminuria stages, in diabetics vs non-diabetics, and across various socioeconomic and geographical groups. The article also presents some data on achieved blood pressure levels and metabolic status in these patients.
The main strengths of the article is the high number of patients included, and the rich description of comorbidities, lab values, blood pressure, and medication classes. Its main weaknesses are the discussion section with little added analytical value beyond description, the unclear presentation with excessively large tables, and no clear statement of how or if consent was obtained from the patients.
Major concerns:
- Discussion section: There is little added analytical value to the discussion beyond descriptions. It would be interesting to hear the authors elaborate further on why treatment rates with ACEi/ARBs were not higher in CKD patients, and in fact lower in albuminuric CKD patients than in albuminuric patients without CKD. Does drug price play a role here? Or the fear of acute kidney injury in an elderly population? Or the low availability of nephrologist care? Statin prescription is more elaborately discussed, but what about the role of statin intolerance/side-effects, drug interactions, "bad publicity" etc in explaining low prescription rates? The discussion section adds little novelty or insight to these issues. Also, possible selection biases are not addressed. For example, documentation of CKD diagnoses is based on doctor-assigned condition codes rather than lab-based eGFR. Furthermore, including only patients with 3 or more GP encounters in the given time frame could possibly introduce selection bias towards older and more multimorbid patients, and a discussion of this, with thoughts on possible non-visiting CKD patients, should be included.
Thank you for your comments. We have now commented on possible reasons for the low rate of RAS inhibitor use in patients with CKD, including non-concordance to treatment guidelines, not referring patients to nephrologist care (page 21, lines 288-290), and fear of inducing acute kidney injury, hypotension, and hyperkalemia in elderly individuals (page 22, lines 295-307). We highlighted that the cost of RAS inhibitors probably had a limited impact on the rate of their prescription as these medications are Pharmaceutical Benefits Schedule (PBS)-subsidised (page 21, lines 290-291).
Internet-driven misinformation may indeed lead patients to quit statin treatment. We have now mentioned that the relatively low rate of statin use in this study could be related to lack of awareness of guidelines and public concerns over perceived adverse reactions following an extensive media campaign about the negative effect of statins (page 23, lines 322-326).
As we detailed in the method section, we included patients with evidence of CKD based on having two estimated glomerular filtration rate (eGFR) or ACR values within three months (page 4, line 92-95). We did not use GP condition codes for the inclusion of patients with CKD in this study, and thus no selection bias related to condition codes was introduced.
Yes, we agree that including only regular patients (who had three or more visits over the previous two years) potentially introduced selection bias by including more patients with CKD and multiple comorbidities (page 25, lines 377-381). We have now mentioned this potential selection bias in our limitation section.
- Missing data: There is no presentation of missing data in the tables or in the text. When some 32,700 patients use antihypertensives out of 44,200 patients, does it mean that almost 12,000 CKD patients use no antihypertensives, or are the data incomplete? Missing data are important in large epidemiological studies, and should be accounted for.
Yes, the numbers are correct. We don’t think there is missing data. Of total 44,259 patients with CKD only 32,782 patients were prescribed at least one antihypertensive medication recorded between February 01, 2016 and June 01, 2016. There were no recorded antihypertensive prescriptions for the remaining 11,477 patients during this period. The most likely explanation is that they were not using this therapy, although it is possible that some patients were obtaining these prescriptions from other general practices (but probably unlikely if they were a regular patient at the practice providing the data). This finding again might relate to the elderly nature of the cohort and a justified concern amongst the doctors over the possibility of inducing hypotension and falls.
- Presentation: Both table 1, 2 and 3 are very long and dense with information. It could perhaps be more informative if table 1 was split into two tables (e.g. 1a with CKD/comorbidities, and 1b with geographical and epidemiological variables). Table 3 could be moved to a supplementary section entirely. Furthermore, Figures 1 and 2 in the Supplementary data would be more readily legible if the PQI variables were tagged with simple key words (such as antihypertensive meds, ACEi/ARB in macroalbuminuria, statin use etc), rather than with the more difficult-to-understand PQI-1, PQI-2 etc. As it stands now, at least table 3 is impenetrable.
Thanks for your comments. We have split ‘Table 1’ into Table 1 (page 8-9) and Table 2 (page 9). We have changed the numbering of subsequent tables accordingly. Although “Table 3”, now numbered Table 4 (page 15-20), is extensive, it contained the definitions of prescribing quality indicators and main findings. We believe that readers can more easily refer to PQIs definitions and results in this table if the table is displayed within the main text.
We have tagged PQI in Figure 1 (page 31) and Figure 2 (page 32) with medication and comorbidities names as suggested.
- Lack of consent: The data were cropped from electronic health records, but there is no proper description of how or if consent was obtained from the patients. I am sure that the study was conducted in an ethically sound way. The paper, however, does not explain what kind of active or assumed consent patients have given to allow their clinical GP data to be published (albeit anonymized).
Yes, details about ethics were provided on page 26 (lines 398-404) of the manuscript. The data we obtained from MedicineInsight for this analysis was de-identified. It did not include patient-specific information (e.g., name, address, date of birth). Therefore, individual consent was not necessary for this study and waived for our ethics application. MedicineInsight informs patients that their de-identified data is used for quality improvement and research through promotional material that is displayed within the waiting rooms of all MedicineInsight participating general practices. Patients’ choice to opt-out from the program was respected, and robust and effective security controls safeguarded their data.
Minor concerns:
- Diabetes vs non-diabetes: Patient demographics and clinical characteristics were presented in a dichotomized way, with diabetics vs non-diabetics. Although biologically interesting, this focus on diabetes vs non-diabetes was not prespecified in the title, abstract or introduction, and does not carry into the discussion. It seems reasonable to either focus more on diabetes vs non-diabetes, or drop it entirely, so that the other scientific points can come to light.
We have now highlighted some important aspects of prescribing quality differences based on patients’ diabetes status in the abstract (page 2 lines 26-28), result (page 8, lines 174-178), and discussion (page 21, lines 280-282).
We added the following: “Antihypertensive medication prescribing was significantly higher in CKD patients with diabetes compared with those without diabetes (82.1% vs. 70.6%, P<0.001). Similarly, 64.1% of CKD patients with diabetes were prescribed an ACEI/ARB compared with 51.5% of those without diabetes (P<0.001). Over 60% of CKD patients with diabetes were prescribed a statin compared with less than 40% without diabetes (P<0.001).”
- Urine tests: The article does not say how many urine tests are analyzed to classify the degree of proteinuria for each patient. KDIGO recommends two or more urine tests. This is not clearly stated here.
We had stated information about urine tests in the method section. “Patients with evidence of CKD based on having two renal function tests that were performed at least three months apart with albumin-to-creatinine ratio (ACR) values ≥3.5 mg/mmol for females or ≥2.5 mg/mmol for males (page 4, lines 94-95)”
- All in all, the paper addresses an interesting topic that is central to the quality of care of the many CKD patients in Australia. However, it is not fully able to present its findings in an easily comprehensible way, or discuss the findings in an analytical way that adds scientific value to the descriptions.
We think the manuscript is now improved and presents the findings in a useful manner. We have added possible reasons for the observed low rate of prescription and some limitations that add scientific value. Details can be seen in our responses above. Thank you.

Round 2
Reviewer 2 Report
The authors have addressed the concerns put forward in my review. They have done so point by point in the covering letter. Also, the revised article is improved, both by a richer discussion section and clearer tables. Finally, the authors have answered the ethical concerns put forward in my earlier review, and added a sentence about it in the paper.
All in all, the article is an interesting and important paper on the quality of CKD care in Australia seen from the general practitioner's point of view.